# Responsive Nanostructured Polymer Particles

**DOI:** 10.3390/polym13020273

**Published:** 2021-01-15

**Authors:** Kang Hee Ku

**Affiliations:** Department of Chemical Engineering and Applied Chemistry, Chungnam National University, Daejeon 34134, Korea; kangheeku@cnu.ac.kr; Tel.: +82-42-821-6696

**Keywords:** block copolymer particles, responsive surfactants, shape-changing particles, polymer nanostructures, stimuli-responsive polymers

## Abstract

Responsive polymer particles with switchable properties are of great importance for designing smart materials in various applications. Recently, the self-assembly of block copolymers (BCPs) and polymer blends within evaporative emulsions has led to advances in the shape-controlled synthesis of polymer particles. Despite extensive recent progress on BCP particles, the responsive shape tuning of BCP particles and their applications have received little attention. This review provides a brief overview of recent approaches to developing non-spherical polymer particles from soft evaporative emulsions based on the physical principles affecting both particle shape and inner structure. Special attention is paid to the stimuli-responsive, shape-changing nanostructured polymer particles, i.e., design of polymers and surfactant pairs, detailed experimental results, and their applications, including the state-of-the-art progress in this field. Finally, the perspectives on current challenges and future directions in this research field are presented, including the development of surfactants with higher reversibility to multiple stimuli and polymers with unique structural functionality, and diversification of polymer architectures.

## 1. Introduction

Responsive polymer particles are designed to alter their properties upon exposure to physical, chemical, and biological stimuli. Shape is one of the most fundamental and essential features that can be imparted on polymeric particles to design their shape-dependent functional behaviors, thus opening up new avenues for providing promising applications in the field of soft materials science. Stimuli-triggered, dynamically shape-switching particles have attracted significant attention due to their tunable rheological behavior, capillary interactions, and optical properties [1,2,3,4,5]. Despite gradual progress in the preparation of responsive polymer particles, most of the examples are micellar structures [6,7,8,9], while the examples of few-micron-sized particles are very limited.

In this regard, solvent evaporation-driven self-assembly of block copolymer (BCP) in emulsions emerges as an ideal method for the creation of soft matter-based colloidal particles with well-defined size, shape, surface patterns, and internal morphologies [10,11,12,13,14,15,16,17,18,19,20,21]. Upon solvent evaporation, the soft and mobile interface of the emulsion leads to a spontaneous deformation of the particle shape, providing a convenient and robust tool for predicting and tailoring the nanostructure of BCP particles. In this process, the precise modulation of the interfacial activity of BCP particles by responsive surfactants is crucial in achieving switchable shape and internal nanostructure of particles in response to external stimuli. The challenge in the design for these shape-transforming BCP particles is to equip appropriate surfactant pairs that can effectively alternate interfacial activity of the particles in response to specific stimuli and to provide the driving force to facilitate the reversible shape transformation.

A number of well-organized reviews covering recent progress in the development of BCP particles have been published in recent years [22,23,24,25,26,27]. Previous papers reviewing BCP particles focused on studies of the internal morphology of BCP particles under the 3D-confinement effect. Several more reviews highlighted the progress on the development of shape-anisotropic BCP particles in terms of physical parameters affecting the final particle shape based on shape control principles. Despite these extensive previous studies, little attention has been paid to the responsive shaping strategies of BCP particles and their applications. In this paper, we focus on the recent advances in the design of responsive polymer particles with well-defined shapes and nanostructures based on the confined assembly of BCPs from emulsions. The remaining important challenges and future directions in this research field are discussed.

## 2. 3D-Confined Assembly of BCPs in Emulsion

### 2.1. Physical Parameters Governing Particle Shape

A key strategy for producing polymer particles, which will be discussed throughout the text, is the self-assembly of multi-component polymers (i.e., BCPs, small molecules, homopolymers, brush polymers, and nanoparticles (NPs)) from solvent-evaporative emulsions. As the organic solvent evaporates from the emulsion, the phase separation of polymer chains into ordered domains occurs from the droplet surface, followed by the propagation of the polymer ordering into the particle center [24,28].

Given that the emulsion containing BCPs acts as a soft and mobile template, the spontaneous deformation of the particle shape is facilitated by bending and stretching of self-assembled polymer chains [29,30,31]. As the microphase-separated structure of the BCPs inside the emulsion has a three-dimensional isotropic symmetry (i.e., spheres (S) or gyroid (G) phases), the resulting particles typically have a spherical shape (Figure 1, left) [32,33]. By contrast, oblate [34,35,36] or prolate [37,38] ellipsoids are formed when BCPs having anisotropic symmetry (i.e., cylinders (C) or lamellae (L)) are confined in emulsions (Figure 1, right). In detail, the total free energy change (ΔG) for the structural development of the BCP particle is expressed as a combination of terms describing (i) the entropic penalty associated with BCP chain stretching upon elongation (ΔG_ent_), (ii) the interfacial energy between two blocks of BCPs (ΔG_int_), (iii) the entropic penalty of bending BCP chains near the edge of the particles (ΔG_ben_), and (iv) the surface energy between the BCPs and the surrounding aqueous medium (ΔG_sur_) [28,39]. This counterbalance of these contributions to minimize the overall free energy determines the final shape of the BCP particles.

### 2.2. Interface Engineering by Dual Surfactant

Control of the interfacial interactions between the BCPs and the surrounding medium is crucial in breaking the symmetry of the interfacial properties of BCP emulsions. Thermodynamically, interfacial interactions are dominated by the surfactants coated on the emulsion surface. Therefore, surfactant engineering can expand the richness of their morphological behavior and shape. At the oil/water interface, surfactants direct the assembly of BCP chains by inducing preferential (or neutral) wetting of either component (or both) [40]. Typically, when a single type of surfactant is used, the parallel orientation of BCPs relative to the surrounding is achieved since most surfactants selectively interact with one of the blocks. To obtain a perpendicular orientation of BCPs, neutralizing the particle/surrounding interaction is necessary. One promising approach is the use of multiple surfactants.

A rational design of dual surfactants for polystyrene-*block*-polybutadiene (PS-*b*-PB) particles has been reported by Jeon et al. [41]. Two different amphiphilic polymeric surfactants, polystyrene-*block*-poly(ethyl oxide) (PS-*b*-PEO) and polybutadiene-*block*-poly(ethyl oxide) (PB-*b*-PEO), favorably interact with PS and PB blocks, respectively. A mixture of PS-*b*-PEO and PB-*b*-PEO in different volume fractions (*f_s_*) was carried out to demonstrate different interfacial contributions at the surface of the symmetric PS-*b*-PB particle (Figure 2a). For the simplest case when *f_s_* = 0 and 1, spherical particles with onion-like nanostructure were formed with a specific block at the outermost layer: PB for *f_s_* = 0 and PS for *f_s_* = 1. On the other hand, various anisotropic particles were generated when mixed surfactants were used, to minimize the surface free energy under the commensurability of the BCPs (i.e., prolate ellipsoids (*f_s_* = 0.46) and tulip-bulb particles (*f_s_* = 0.36)). A similar design was reported by Klinger et al., by using cetyl trimethyl ammonium bromide (CTAB) and 16-hydroxy-*N*,*N*,*N*-trimethylhexadecan-1-ammonium bromide (HO-CTAB) as a complementary set of dual surfactants for symmetric polystyrene-*block*-poly(2-vinyl pyridine) (PS-*b*-P2VP) [39]. An intermediate ratio of CTAB and HO-CTAB led to the generation of ellipsoidal PS-*b*-P2VP particles with a striped nanostructure.

Furthermore, the versatility of synthesis in the precise adjustment of NP size, shape, and surface chemistry highlights the superior potential of NP surfactants over traditional organic surfactant molecules [31,47]. For example, when polymer-coated Au NPs segregate to the three-phase interface comprised of PS, P2VP, and the surrounding medium, they induce a dramatic shift in the internal morphology and overall shape of the PS-*b*-P2VP particles from spheres to ellipsoids with axially stacked lamellae [37]. The positioning of NPs at the surface of the BCP particles can be further controlled in terms of their size and shape, making NP surfactants more powerful in the engineering of the emulsion property. The relative size-and-length-ratio of NPs (*d* or *l*) over the NP-hosting domain (*L*), that is, *d*/*L* (or *l*/*L*), is a critical parameter for determining their location and ability to function as surfactants [42,48,49,50]. When length-controlled CuPt NPs (i.e., 2.3 nm < *l* < 50 nm) were used to modulate the surface of the polystyrene-*block*-poly(4-vinyl pyridine) (3-pentadecylphenol) (PS-*b*-P4VP(PDP)) particles, the oblate particles were generated only in the range of 0.36 ≤ *l*/*L* ≤ 0.96, whereas the prolate particles were produced for a much wider range of *l*/*L* ≥ 0.83 without an upper limit (Figure 2b) [50]. Moreover, surfactant efficiency was greatly amplified by using chemically designed graphene quantum dots (GQDs), which have a plate-like shape. The addition of alkyl ligand-grafted GQDs to the cylinder-forming PS-*b*-P4VP(PDP) particle resulted in a dramatic transition from conventional spherical shape to oblate particles.

Another unique system neutralizes surface interaction of triblock copolymers (triBCPs) to generate non-spherical particles with three-phase domains (Figure 2c). For example, a mixture of CTAB and poly(vinyl alcohol) (PVA) provides a neutral interaction for PS-*b*-PI-*b*-P2VP triBCPs (PI: polyisoprene), affording the formation of prolate shape [43]. Confinement assembly of ABC triblock terpolymers has been further reported by varying the type of each block: PS-*b*-PB-*b*-PMMA, PS-*b*-PB-*b*-PtBMA, and PS-*b*-P4VP-*b*-PtBMA (PMMA: poly(methyl methacrylate), PtBMA: poly(t-butyl methacrylate)) [44,51]. Under spherical confinement stabilized by CTAB, the PS-*b*-PB-*b*-PtBMA triblock terpolymer adopts a hemispherical shape with a mixture of concentric lamella–axial lamella morphology [44]. Cross-linking and disassembly of the microparticles further resulted in well-defined nanorings or Janus nanocups with different chemistries on the inside and outside. While microphase diagrams provide a fairly extensive opportunity to search for morphologies found only in ABC triblock polymers, it still remains a great challenge for the synthetic screening of all possible block compositions.

Blending another incompatible homopolymer on a BCP-containing oil-in-water emulsion system is a feasible way to achieve additional anisotropy of the particle. The interplay of macrophase separation between BCPs and homopolymers and microphase separation of BCPs determines the final shape and nanostructure of the resulting particle. Thermodynamically, additional macrophase separation generates another polymer–polymer interface between BCPs and homopolymers that can affect the internal nanostructure and shape by changing the interfacial energy between each polymer component. For example, unique conical Janus particles with hierarchical nanostructure were reported for blends of PS-*b*-P4VP and PMMA (Figure 2d) [45] and blends of PS-*b*-PB and poly(methylmethacrylate-statistical-(4-acryloylbenzophenone)), respectively [52]. The key is properly adjusting the incompatibility between A-*b*-B and C polymers and the neutrality condition of BCPs and their surroundings.

Finally, the morphological behavior of BCPs under strong 3D confinement within emulsion depends on the magnified interfacial interaction between BCP particles and the surrounding medium and the stretching/bending penalty of polymer chains. Therefore, BCPs assembled under strong confinement, typically when the particle size is less than 100 nm (D/L_0_ < 4.0), can generate unconventionally structured particles with various shapes and internal structures. In this range of particle size, the diameter of the particle becomes comparable to the periodicity of the BCPs. Based on these principles, the fabrication of patchy particles with a variety of 3D shapes was demonstrated by exploiting PS-*b*-P4VP having high molecular weights (Figure 2e) [29,53]. Depending on the volume of the particle, a series of patchy particles of snowmen, dumbbells, triangles, tetrahedra, and raspberries were prepared [46].

Therefore, surfactants not only act as stabilizers for emulsion droplets containing BCPs but also rearrange themselves by dynamic adsorption and desorption to minimize surface area and support the internal structure during solvent evaporation, which gives rise to the deformation of emulsions to anisotropic shapes.

## 3. Stimuli-Responsive BCP Particles

Spatial and temporal control over the interfacial properties of particles can be achieved using surfactants that respond to a variety of triggers, including light, temperature, pH, redox, and magnetic field [54,55,56,57,58]. Several examples of responsive monomeric blocks are illustrated in Figure 3 [59]. Given that the rearrangement is driven by molecular entities capable of responsiveness to stimuli in a given environment, it is necessary to provide suitable molecular building blocks that can respond to polymeric solutions to exhibit adjustable stimuli-responsiveness. In particular, BCP assemblies can undergo morphological transformations through the solvent annealing process, where the chain rearrangement is caused by absorbing solvents. During this process, the use of responsive polymers at the interface not only allows the interface properties to be adjusted according to the environment but also allows the corresponding shape-switchable properties to be incorporated into the BCP particles.

### 3.1. Temperature

A well-known polymer with lower critical solution temperature (LCST) behavior is poly(N-isopropylacrylamide) (PNIPAM), which exhibits the coil-to-globule phase transition at 32 °C [60,61]. Lee et al. reported temperature-driven shape transformation of PS-*b*-P4VP particles by utilizing PNIPAM surfactants (Figure 4) [62]. In this system, PNIPAM stabilized the oil-in-water emulsions as a P4VP-selective surfactant, creating a near-neutral interface between the PS and P4VP domains with CTAB, and anisotropic PS-*b*-P4VP particles (i.e., prolate and oblate particles) were realized. Importantly, the temperature-directed arrangement of PNIPAM according to its solubility determined the overall shape of the BCP particles. Prolate particles were produced above the critical temperature, whereas oblate particles were obtained below the critical temperature. The temperature window of the particle shape transition was widely adjustable by developing other PNIPAM derivatives with different LCST values (i.e., poly(*N*-*n*-propylacrylamide) (PNNPAM) and poly(*N*-isopropylmethacrylamide) (PNIPMAM)).

More recently, this system was further improved by exploiting pH- and thermal-responsive poly(dimethylaminoethylmethacrylate-*random*-*N*-isopropylacrylamide) (poly(DEAEAM-*r*-NIPAM)) surfactants to achieve pH and temperature dual-responsive PS-*b*-P4VP particles in very narrow ranges (around pH 6.5 and 40 °C) [63].

### 3.2. pH

A typical structure of a pH-sensitive polymer involves ionizable groups that undergo reversible ionization at different pH values, causing pH-dependent swelling/deswelling behavior. In this case, the hydrophobic volume along the polymer chain may alternately extend or collapse due to electrostatic repulsions between generated charges [59,64]. Klinger et al. reported prolate PS-*b*-P2VP particles, where the cross-linked hydrogel P2VP layers enabled a reversible anisotropic shape transformation of the whole ellipsoidal particles (more than twice along the major axis) in response to a pH change (Figure 5) [39].

Similar behavior was observed in the system of PS-*b*-PI-*b*-P2VP triBCP particles [20]. In addition, the selective introduction of benzophenone as a photocrosslinking group produced particles with reversible shape changes induced by triggered swelling/deswelling where the dynamic behavior was additionally combined with other functionalities, such as ferrocene groups or reactive pentafluorostyrene moieties [38]. For example, the ferrocene-containing PFS block rendered these particles redox-responsive. By adding an oxidant (i.e., FeCl_3_), a shape transition of the particles was observed due to a change of PFS polarity from hydrophobic to hydrophilic upon oxidation. The combination of redox-responsive PFS with pH-responsive P2VP domains enabled a multi-stimuli-responsive behavior, which represents a powerful platform for a wide range of applications. Very recently, the use of pH-responsive core-cross-linked NPs to facilitate shape transformation of BCP particles has been successfully demonstrated by exploiting polystyrene-*block*-poly(dimethylsiloxane) (PS-*b*-PDMS) and cross-linked PS-*b*-P4VP NPs [65]. The PS-*b*-PDMS microparticles transformed from elongated Janus pupa-like particles to onion-like particles by decreasing the pH value of the aqueous phase.

### 3.3. Light

If the surfactant molecule contains a suitable chromophore, light irradiation can yield different physical photo-induced responses. Distinct from other physicochemical stimuli (i.e., pH, temperature, redox, and biomolecules), light-responsiveness offers powerful capabilities as a trigger stimulus because of its superior spatial and temporal resolution with negligible time delay [66,67,68,69]. Moreover, selective light-responsive behavior can be obtained by modulating the wavelength and intensity of light, which provides a powerful strategy for tuning particle shape in a programmed manner [70,71,72,73]. In the pursuit of light-responsive systems, a tremendous amount of research efforts have been made, including the development of photo-responsive polymers and corresponding assemblies by introducing photodegradable or photochromic units that undergo reversible isomerization upon light irradiation [74,75,76,77,78,79,80,81,82,83,84].

Light-responsive shape-changing PS-*b*-P2VP particles (i.e., from onion to prolate or oblate ellipsoids) were developed by using surfactants containing photocleavable nitrobenzyl ester or coumarin ester groups by Kim’s group (Figure 6) [85]. The cleavage of surfactants induced sequential modulation of the amphiphilicity and interfacial activity of the surfactants (i.e., interaction between surfactants and each block), leading to the modification of the surface and wetting properties of BCP particles. The use of a mixture of two photo-responsive surfactants that exhibit different activation wavelengths (i.e., 254 and 420 nm) afforded wavelength-selective shape transformations of the BCP particles. Adding reversible properties to photoactive surfactants (i.e., photochromic units that can be reversibly isomerized between different states when exposed to light) would result in completely reversible and dynamic changes in particle shape and properties, greatly expanding its usefulness for other practical applications.

### 3.4. Solvent and External Fields

Utilizing BCPs as a single particle Bragg reflector is another emerging application area. As the BCPs possess a sufficient periodic domain size, their internal nanostructures are potentially applicable as photonic crystals whose structural color depends on the thickness and the number of periodic layers [86]. Considering Bragg’s equation, for bright reflections in the visible light region, the BCP periodicity should be larger than 120 nm, and the number of nanostructured photonic layers should exceed 40 [87,88]. It remains a challenge to extend the current strategy of BCP assembly to ultra-high molecular weight building blocks with regularly ordered structures greater than a few tens of micrometers in diameter, which is particularly important in optical applications. As an alternative, the addition of molecular additives, swelling of the entire domain promoted by the solvent, and utilization of brush block copolymers have been attempted.

Zhu et al. reported PS-*b*-P2VP photonic particles with high molecular weight (i.e., higher than 250 k) with concentric lamellae. The degree of solvent swelling was controlled by the density of cross-linking. Selective swelling of P2VP domains by the solvent (ethanol) enabled the reflective color to shift from blue to red as a function of cross-linking density (Figure 7a) [89]. Similarly, responsive photonic crystal microcapsules of PS-*b*-P2VP have been developed from a water-in-oil-in-water double emulsion [90]. Compared to photonic particles, the microcapsules exhibited a bright structural color with significantly enhanced monochromaticity due to the absence of irregular cores. The structural color of microcapsules showed sensitive responsiveness to pH value and evolved from blue to red when the pH value decreased from 6.8 to 3.5 (Figure 7b). The corresponding reflection peak showed a red-shift from 422 to 650 nm.

An external field, such as an electric field and magnetic field, can be applied to manipulate the orientation and movement of particles in the desired manner. Recently, photonic ellipsoids functionalized with magnetic NPs were developed by the Swager group to explore a real-time on/off coloration activated by the magnetic field (Figure 8) [91]. The photonic ellipsoids were designed by use of dendronized brush block copolymers composed of alkyl wedge groups and benzyl wedge groups (poly(AW-*b*-BnW) den-BBCP), where the full-color reflection was achieved by tuning overall molecular weights. Surfactants composed of dendritic monomer units enabled precise modulation of the interfacial properties of the polymer particles from spheres to ellipsoids. Due to the shape anisotropy of the ellipsoids, the photonic behavior of the prolate particles with axially-stacked lamellae was strongly dependent on the angle of incident light to the photonic layers, which cannot be observed in spherical photonic particles with concentric lamellae. The magnetic field-assisted orientation-dependent photonic behavior was explored, achieving a real-time switchable on/off color response.

## 4. Conclusions and Perspectives

To date, remarkable progress has been made in the field of confinement self-assembly of BCPs from evaporative emulsions. Various strategies for modulating interface properties have successfully led to a variety of non-spherical particle libraries with controlled internal morphologies and overall shapes. Importantly, surfactants play a critical role in stabilizing the emulsion as well as directing the assembly of the polymer by determining the preferential surface wetting of the BCP component. In this review, we highlighted recent advances in shape-controlled polymer particles, especially for their responsive function, which is summarized in Table 1.

While stimuli-responsive surfactants have received a great deal of attention due to their tunable chemical, physical, and other properties in response to a variety of external stimuli, the manufacturing of dynamic BCP particles that uses responsive surfactants to control their shape and structure remains relatively unexplored. Systematic research on the development of surfactants is required to create smart particles that respond highly reversibly to multiple stimuli without any fatigue during multiple cycles. We also note that most responsive polymer particles require solvent-mediated reconstruction to provide sufficient chain mobility. The use of polymers having a low glass transition temperature (T_g_) or cross-linkable units will be a promising alternative approach. Moreover, although some research on non-linear polymer assemblies has been investigated recently, there remains a need for polymer particle design principles for new architectures (Figure 9). Polymer architecture plays a significant role in controlling the assembly structure as well as its responsive behavior. Spiral polymers and gyroid polymers, for example, are two of the promising prospects for designing directional release systems in response to external stimuli. Finally, the unique structural functionality, i.e., biodegradable BCPs, can be expanded by modifying their compositions.

## Figures and Tables

**Figure 1 polymers-13-00273-f001:**
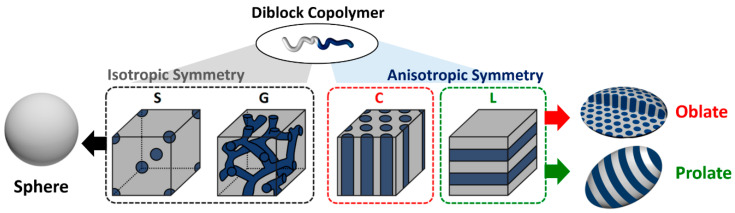
Schematic illustrations of the block copolymer particle (BCP) phases in bulk (S: spheres, G: gyroid, C: cylinders, and L: lamellae) and the corresponding structure of polymer particles at neutral conditions (reprinted with permission from [28], Copyright 2019, American Chemical Society).

**Figure 2 polymers-13-00273-f002:**
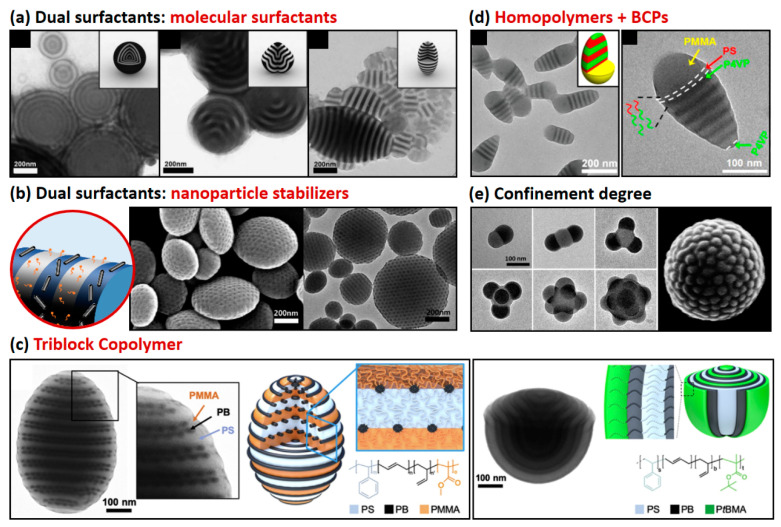
Interfacial engineering of BCP particles to produce non-spherical BCP particles by surface neutralization. (**a**) Transmission electron microscopy (TEM) images of symmetric polystyrene-block-polybutadiene (PS-*b*-PB) prepared by mixed surfactants of polystyrene-block-poly(ethyl oxide) (PS-PEO) and polybutadiene-block-poly(ethyl oxide) (PB-PEO) at different mixing ratios: spherical, tulip-bulb, and prolate particles (reprinted with permission from [41], Copyright 2008, Wiley-VCH). (**b**) Scanning electron microscopy (SEM) and TEM images of oblate polystyrene-block-poly(4-vinyl pyridine) (3-pentadecylphenol) (PS-*b*-P4VP (PDP)) particles prepared by a mixture of cetyl trimethyl ammonium bromide (CTAB) and oleic acid-coated Au NPs (reprinted with permission from [42], Copyright 2014, American Chemical Society). (**c**) TEM images of prolate PS-*b*-PB-*b*-PMMA triBCP (PMMA: poly(methyl methacrylate; triBCPs: triblock copolymer) particles stabilized by the mixture of CTAB and poly(vinyl alcohol) (PVA) and tulip-bulb-like particles of PS-*b*-PB-*b*-PtBMA triBCPs (PtBMA: poly(t-butyl methacrylate)) stabilized by CTAB as a sole surfactant (reprinted with permission from [43], Copyright 2019, American Chemical Society, and [44], Copyright 2019, Wiley-VCH). (**d**) TEM images of conical Janus particles comprised of PS-*b*-P4VP and PMMA bends (reprinted with permission from [45], Copyright 2014, American Chemical Society). (**e**) TEM images of PS-*b*-P4VP patchy particles with a tunable number of bulbs and SEM image of raspberry-like PS-*b*-P4VP particles (reprinted with permission from [29,46], Copyright 2015 and 2012, American Chemical Society).

**Figure 3 polymers-13-00273-f003:**
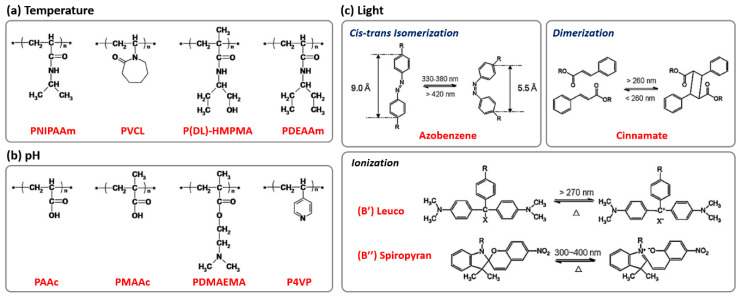
Examples of molecular structures responsive to (**a**) temperature, (**b**) pH, and (**c**) light (reprinted with permission from [59], Copyright 2010, Elsevier).

**Figure 4 polymers-13-00273-f004:**
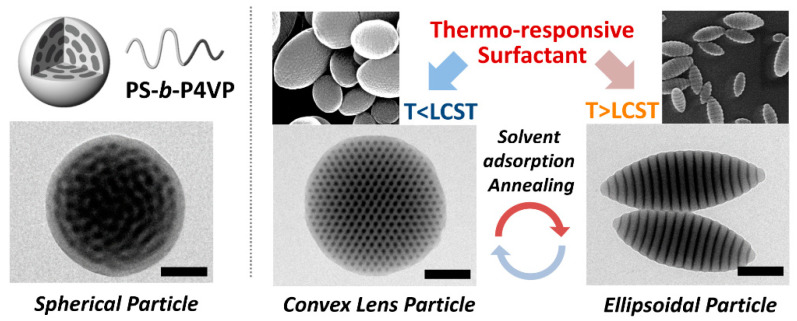
Temperature-driven transformation of the shape and morphology of PS-*b*-P4VP particles using temperature-responsive poly(N-isopropylacrylamide) (PNIPAM) surfactants (reprinted with permission from [62], Copyright 2017, Wiley-VCH).

**Figure 5 polymers-13-00273-f005:**
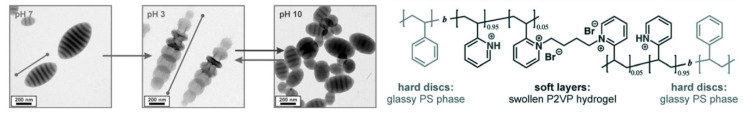
pH-triggered dynamic shape changes for PS-*b*-P2VP particles with cross-linked P2VP domains as a result of swelling/deswelling of the P2VP-based hydrogel discs (reprinted with permission from [39], Copyright 2014, Wiley-VCH).

**Figure 6 polymers-13-00273-f006:**
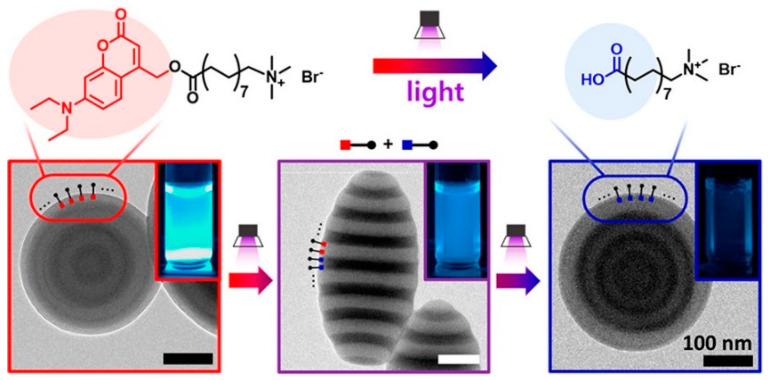
Light-responsive shape and color transformation of PS-*b*-P2VP particles prepared with surfactants having coumarin ester photo-cleavable groups (reprinted with permission from [85], Copyright 2019, American Chemical Society).

**Figure 7 polymers-13-00273-f007:**
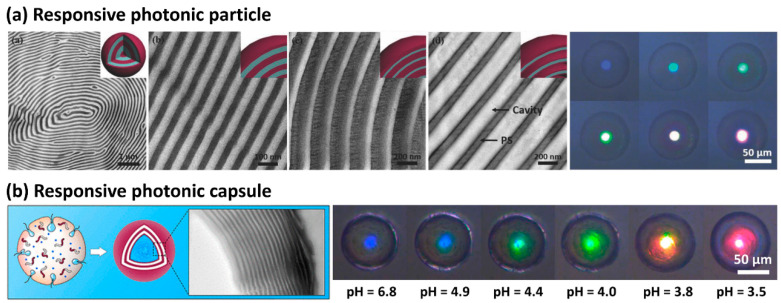
(**a**) Solvent-responsive photonic particles of PS-*b*-P2VP comprising concentric lamellae (reprinted with permission from [89], Copyright 2018, Wiley-VCH). (**b**) pH-responsive photonic crystal microcapsules of PS-*b*-P2VP with parallel lamellar layers (reprinted with permission from [90], Copyright 2020, American Chemical Society).

**Figure 8 polymers-13-00273-f008:**
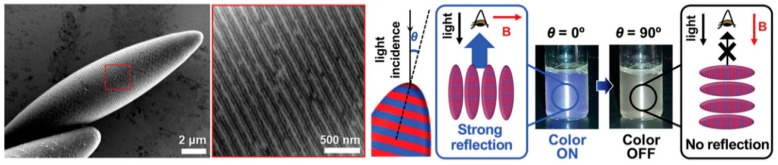
Magnetic-responsive color-switchable photonic ellipsoidal particles created from dendronized brush block copolymers (reprinted with permission from [91], Copyright 2020, American Chemical Society).

**Figure 9 polymers-13-00273-f009:**
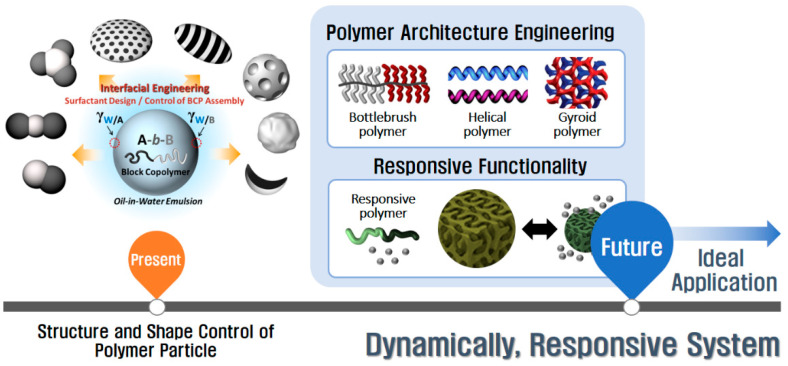
Remaining important challenges and directions for dynamically responsive nanostructured polymer particles.

**Table 1 polymers-13-00273-t001:** Summary of responsive BCP particles.

Stimulus	BCP	Surfactants	Ref.
Temperature	PS-*b*-P4VP	poly(*N*-isopropylacrylamide)	[62]
		poly(DEAEAM-*r*-NIPAM) (dual)	[63]
pH	PS-*b*-P4VP	poly(DEAEAM-*r*-NIPAM) (dual)	[63]
	PS-*b*-P2VP	CTAB, CTAB-OH	[39]
	PFS-*b*-P2VP	CTAB, CTAB-OH	[38]
Light	PS-*b*-P2VP	Coumarin ester-CTAB, Nitrobenzyl ester-CTAB	[85]
Solvent	PS-*b*-P2VP	PVA	[89,90]
Field	poly(AW-*b*-BnW) den-BBCP	AW-CTAB, BnW-CTAB, CTAB	[91]

## Data Availability

The data presented in this study are available on request from the corresponding author.

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
