# Peer review of "Responsive Nanostructured Polymer Particles"

_polymers, 2021, doi:10.3390/polym13020273_

Round 1

Reviewer 1 Report

The manuscript entitles by Kang Hee Ku entitled “Responsive Nanostructured Polymer Particles” is based on an exciting and emerging polymer science area which has recently been reviewed by many research groups around the world but yet the elusiveness of the shaping tuning and application aspects of co-block polymers is evident. Although the overall review is generally well organised and written, however, I feel the following points should be carefully considered before this could be accepted.

Major:

1- Abstract require a line on the overall aim of your review and why you think it is essential for the scientific community

2- Line 16-17, what these future directions and challenges are, please concisely write these here.

3- In introduction rationale of current review require to revisit, please can author precisely elaborate and add how various researchers approach this topic? Furthermore, where exactly the grey area is which the author wants to cover. In the current stage, this section is slightly weak and require attention

4- In rationale, it would be interesting to add, on what basis you have selected these studies, have you used any search databases or manual searching? Key searching words being used?

Minor:

1- Line 8 “responsive particles” I think it should be “responsive polymer particles”

2- Figure 3 should be after the text, I am not sure how we can start a section with a Figure?

3- Section 3.4, “OTHERS” on what basis this classification has been established? I think it would good if you start this section with clarifying about the section for readers to establish what is coming up.

4- I think it would be a good idea to tabulate all the key findings to the included studies, as this will enhance the viewership of this manuscript.

5- Section 5, “PATENTS” this is empty, please either remove or add the section findings.

Reviewer 2 Report

Shape-controlled synthesis of polymeric particles is of fundamental importance in designing structured materials. Author has written an interesting and well balanced incremental progress report (rather than a full review). A major problem with this kind of papers is their current saturation in the literature. This manuscript, however, covers some sections that have not been often considered and, moreover, the quality of the graphical support is good. Overall, it's a good publication.

In summary, while much of this report is excellent, more useful would be to use the platform created by the author to critically address specifically where the future progress in this field is expected. 

Round 2

Reviewer 1 Report

My suggestions have been adopted by the author, so now this report may be considered for acceptance.